# Zero-Shot Non-Autoregressive TTS Beyond Autoregressive Models via Soft Alignment Generation and Residual Modeling

## Abstract

Autoregressive TTS leverages soft alignment generated by the attention mechanism, which provides the decoder with a well-designed context vector. Subsequently, the decoder receives both the semantic representation and the acoustic representation generated at the previous time step. For this reason, autoregressive TTS achieves strong performance. Thus, we propose novel algorithms to bring similar benefits to non-autoregressive TTS. First, we propose a method to distill soft alignments—originally provided by attention in autoregressive models—into a flow matching model trained between mel-spectrograms and text representations. This allows non-autoregressive models to leverage attention-like context vectors without requiring autoregressive decoding. Second, we introduce an invertible encoder, designed based on normalizing flow, to disentangle semantic and residual acoustic representations. The invertible encoder maps the residual information, which is absent in the context vector, closer to a Gaussian distribution. During inference, we can treat the context vector as the semantic representation and Gaussian noise as the acoustic representation. Lastly, to improve zero-shot TTS performance, we propose a prompt-aware lightweight convolution, where the kernel weights are dynamically adjusted for each speech prompt. With the proposed methods, our non-autoregressive TTS model achieves comparable performance to existing autoregressive models.

## 1 Introduction

Zero-shot text-to-speech (TTS) Casanova et al. (2022); Wang et al. (2023) aims to synthesize speech that reflects the voice characteristics of unseen speakers at inference time, without requiring any fine-tuning. Recent advances in large language models (LLMs), such as GPT Achiam et al. (2023) and T5 Ao et al. (2021), have inspired TTS architectures to adopt similar transformer-based models and in-context learning strategies. In zero-shot TTS, speaker-specific information is typically captured from a short speech prompt (e.g., 3-second mel-spectrogram), enabling generalization to unseen speakers. This prompt-based paradigm has shown significant performance improvements over traditional speaker embedding approaches. For instance, VALL-E Wang et al. (2023) leverages a decoder-only transformer and 60K hours of training data to achieve strong zero-shot performance via in-context learning. Most state-of-the-art zero-shot TTS systems Kim et al. (2024); Lee et al. (2024) adopt autoregressive (AR) Wang et al. (2017); Shen et al. (2018) architectures due to their ability to model temporal dependencies and leverage rich contextual information. These models generate acoustic features (e.g., mel-spectrograms or codec tokens Défossez et al. (2022)) sequentially, using previously generated outputs as inputs for subsequent steps. Additionally, attention mechanisms provide soft alignment between text and acoustic features, allowing the decoder to access a fine-grained context vector that has rich context information. Consequently, autoregressive TTS systems typically outperform non-autoregressive (NAR) Ren et al. (2019; 2020) models in terms of speech quality. Since the AR TTS systems Neekhara et al. (2024); Battenberg et al. (2024); Kim et al. (2024) try to overcome the instability or low latency of AR, the performance of AR becomes higher.

On the other hands, apart from the general problems arising from the aforementioned AR, NAR models tend to lag behind AR models in performance, mainly due to two reasons: (1) NAR models

Figure 1: The concepts of our proposed method that supplements missing acoustic information in the semantic representation and refines the existing upsampling process using soft alignment generation.

lack access to previously generated acoustic features during decoding, and (2) most NAR models rely on hard and shallow upsampling techniques (e.g., duration-based duplication) rather than soft, flexible alignment mechanisms. To bridge this gap, as shown in Figure 1, we introduce **Re**sidual modeling and **so**ft alignment generation-based NAR-**TT**S that can be **o**n par with performance of AR (RisoTTo) as follows:

**Soft Alignment Generation (SAG):** In autoregressive TTS, attention mechanisms generate soft alignments between text and acoustic features, allowing the decoder to condition on fine-grained contextual information. However, non-autoregressive models cannot use such attention-based alignment, as acoustic frames are generated in parallel. Instead, they typically rely on hard and shallow upsampling methods. To address this limitation, we introduce SAG, which employs flow matching Lipman et al. (2022), which uses only text representation to generate soft alignments between the mel-spectrogram and the text representation. This enables the model to leverage alignment information similar to that of autoregressive attention, enhancing contextual richness during inference.

**Invertible Encoder (IE):** The invertible encoder disentangles acoustic and semantic information by modeling the residual component between the mel-spectrogram and the context vector. Specifically, it maps the residual acoustic features—those not captured by the semantic representation—into a Gaussian distribution using a normalizing flow. At inference time, acoustic information can be sampled from this distribution, complementing the semantic context and improving synthesis quality.

**Prompt-Aware Lightweight Convolution (PAL):** Inspired by SC-CNN Yoon et al. (2023), which improves zero-shot TTS by generating convolutional kernel weights from speaker embeddings, we adopt a lightweight convolutional module whose kernel weights are directly extracted from the speech prompt. This enables prompt-adaptive feature modulation and enhances generalization to unseen speakers in zero-shot settings.

## 2 BACKGROUND

### 2.1 FLOW MATCHING WITH OT PATH

Flow matching Lipman et al. (2022) estimates a probability path between data $x_1$ and prior $x_0$ distributions. Lipman et al. (2022) defines optimal transport (OT) path based on Gaussian conditional probability path that forms a straight trajectory between $x_0$ and $x_1$. The OT path on time step $t \in [0, 1]$ varies depending on $\mu_t$ and $\sigma_t$, which can be defined as follows: $\mu_t = tx_1$ and $\sigma_t = 1 - (1 - \sigma_{min})t$. Since a probability distribution on the OT path follows a Gaussian distribution, $x_t$ on time step $t$ can be computed by affine transform as follows: $x_t = tx_1 + (1 - (1 - \sigma_{min})t)x_0$. Consequently, the vector field $u_t$, which generates a desired probability path, is defined via the ordinary differential equation as follows:

$$\frac{dx_t}{dt} = u_t = x_1 - (1 - \sigma_{min})x_0. \tag{1}$$

Flow matching is trained to predict the vector field $u_t$ corresponding to $x_t$ using mean squared error (MSE) loss function between predicted vector field $v_t$ and the target $u_t$. Thus, flow matching can generate data from prior distribution by repeating the following equation until the time step $t$ becomes 1 from 0:

$$x_{t+dt} = x_t + v_t dt, \tag{2}$$

where $dt$ is set to $\frac{1}{N}$ and $N$ is the total number of sampling.

## 2.2 INVERTIBLE ENCODER

A Normalizing flow (NF) network is a type of generative model that uses an inverse function of flow to generate data. Inspired by Rombach et al. (2020), we construct an NF network based on decoder of GlowTTS Kim et al. (2020) to extract the latent variable $z$ from the conditional NF network, as illustrated in Figure 2. This conditional NF network takes two inputs: $x$ and $c$, to generate a conditional data distribution $p(x|c)$ that is normalized to the prior distribution. The log-likelihood of the data distribution $p(x|c)$ is calculated as follows:

$$\log p(x|c) = \log p(z|c) + \log |\det(\frac{\partial f(x)}{\partial x})|, \tag{3}$$

where $p(z|c)$ represents the output of the NF network. To train the NF network, the negative log-likelihood $-\log p(x|c)$ is decomposed into Kullback-Leibler (KL) divergence and entropy as follows:

$$\text{KL}(p(z|c)|q(z)) + H(x|c), \tag{4}$$

where $q(z)$ is the prior distribution (typically standard Gaussian), and $H(x|c)$ denotes the constant data entropy. According to Alemi et al. (2016), minimizing $\text{KL}(p(z|c)|q(z))$ reduces the mutual information between $z$ and $c$, effectively disentangling $z$ from $c$. This allows us to extract residual information $z$ from $x$, independently of $c$, since $q(z)$ is selected independently of $c$. The mutual information $I(z, c)$ between $z$ and $c$ is represented by:

$$I(z,c) = \int p(z,c) \log \frac{p(z,c)}{p(z)p(c)} = \int p(z,c) \log \frac{p(z|c)}{p(z)} = \int p(z,c) \log p(z|c) - \int p(z) \log p(z)$$

$$\leq \int p(z,c) \log \frac{p(z|c)}{q(z)} = KL(p(z|c)\|q(z)). \tag{5}$$

## 3 METHOD

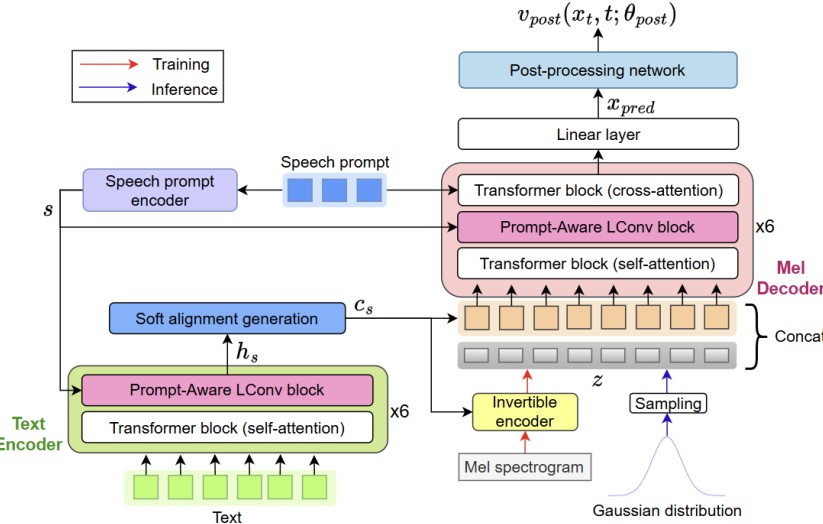

Figure 2: Overall architecture of RisoTTo.

We propose methods to improve the performance of zero-shot non-autoregressive text-to-speech (TTS) systems. Unlike autoregressive TTS models, non-autoregressive ones face difficulties in generating soft alignments between text and acoustic features due to the absence of autoregressive attention mechanisms and acoustic context during inference. As a result, they typically rely on hard upsampling methods, such as Gaussian upsampling Donahue et al. (2020), which limits expressiveness and accuracy.

To address this limitation, we introduce a *Soft Alignment Generation (SAG)* network based on flow matching that learns to produce soft alignments between text and mel-spectrograms without access

to acoustic features during inference. This allows for more flexible and semantically enriched context modeling. Additionally, to compensate for the lack of acoustic information, we propose an *invertible encoder* that disentangles residual acoustic features from the mel-spectrogram and maps them to a Gaussian distribution. By sampling from this distribution during inference, we can reintroduce acoustic context, bridging the gap between non-autoregressive and autoregressive decoding. Finally, to improve speaker adaptation, we propose a prompt-aware lightweight convolution (LConv) block, which uses speech prompts to modulate the model dynamically in a zero-shot setting.

### 3.1 Model description

As shown in Figure 2, our non-autoregressive TTS model consists of several modules. The speech prompt encoder receives a speech prompt, which is a randomly segmented 3-second mel-spectrogram extracted from the reference speech, and encodes it into a fixed-size vector $s$ representing speaker characteristics. This vector $s$ is then used in the prompt-aware LConv block, which integrates both text and speaker information. Consequently, the text encoder extracts a speaker-dependent text representation $h_s$. The representation $h_s$ is upsampled by the Soft Alignment Generation (SAG) network, which comprises a Conv2D-UNet-based flow matching network and an attention mechanism. The upsampled $h_s$ using SAG becomes a context vector $c_s$, which has the same length as the mel-spectrogram. The context vector $c_s$ serves as a conditional feature for the invertible encoder, which takes the mel-spectrogram as input. Invertible encoder extracts residual information between $c_s$ and mel-spectrogram, and mel decoder predicts the mel-spectrogram conditioned on both the context vector and residual information. Finally, the predicted mel-spectrogram is refined to a higher-quality output using a flow matching-based post-processing network.

**Text encoder** comprises 6 Transformer blocks and 6 PAL blocks. Both the input and output dimensions of the Transformer and PAL blocks are set to 256. Text encoder is conditioned by $s$ and extracts speaker-dependent text representation $h_s$

**Mel decoder** adopts the same PAL and self-attention-based Transformer block configuration as the text encoder, while additionally incorporating a cross-attention–based Transformer to extract information from the speech prompt. A linear projection layer in the mel decoder maps the 256-dimensional hidden representations to the 80-dimensional mel-spectrogram space. The predicted mel-spectrogram $x_{pred}$ is then compared to the target mel-spectrogram $x_{target}$ using the $l2$ loss as follows:

$$\mathcal{L}_{\text{mel}} = ||x_{target} - x_{pred}||_2^2. \tag{6}$$

**Post-Processing network (PostNet)** $v_{post}$ is based on flow matching conditioned on a prior distribution, which is obtained by adding Gaussian noise $\epsilon$ sampled from $N(0, I)$ to the predicted mel-spectrogram $x_{pred}$. This results in a prior distribution defined as $N(x_{pred}, I)$. The vector field for the flow matching-based PostNet is then formulated based on this prior as follows:

$$u_t^{post} = x_{target} - (1 - \sigma_{\min})(x_{pred} + \epsilon). \tag{7}$$

$v_{post}$ is trained by $\mathcal{L}_{\text{post}}$ as follows:

$$\mathcal{L}_{\text{post}} = ||u_t^{post} - v_{post}(x_t, t; \theta_{post})||_2^2, \tag{8}$$

where $\theta_{post}$ is learnable parameters, and PostNet adopts the same Conv1D-UNet-based flow matching architecture as employed in Matcha-TTS Mehta et al. (2024). To extract speaker information, we utilize a speech prompt encoder that generates a fixed-size embedding, following the reference encoder design of Grad-TTS Popov et al. (2021). As discussed previously, such fixed-size representations may be less effective than directly leveraging the speech prompt for in-context learning. To address this limitation, we further incorporate in-context learning by using the speech prompt within the mel decoder. Additional details regarding other modules are provided in following sections.

### 3.2 Soft Alignment Generation

As shown in Figure 3(a), soft Alignment Generation (SAG) consists of an attention mechanism, a Conv2D-UNet-based flow matching network $v_{SAG}$, and a duration predictor. The attention mechanism computes the matrix multiplication between the speaker-dependent text representation $h_s$ and

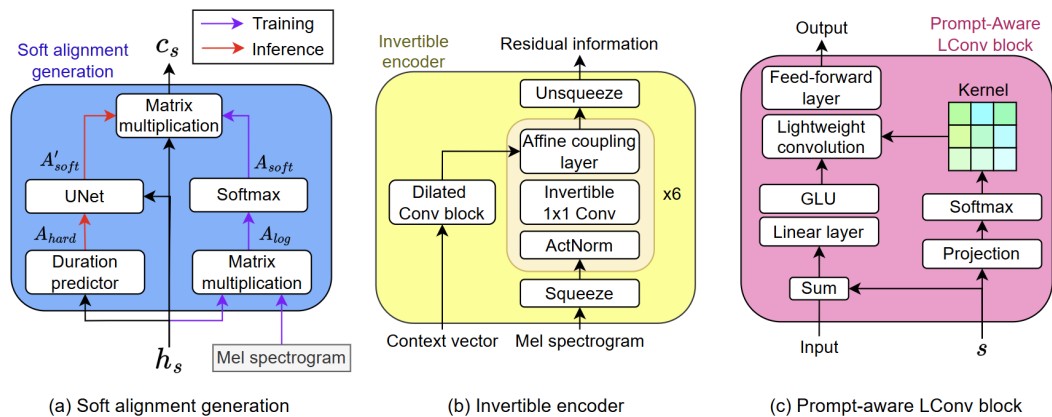

Figure 3: Architectures of (a) soft alignment generation network, (b) invertible encoder, and (c) prompt-aware lightweight convolution. In (a), $\tilde{A}_{log}$ and white dotted box denote reconstructed $A_{log}$ and attention mechanism, respectively. In (b), the residual information is projected from 80 to 256 dimensions to enable summation with the context vector.

the mel-spectrogram, providing a soft alignment that maximizes the speech likelihood of the TTS model, such as autoregressive TTS models:

$$\text{Attention}(h_s, x_{target}) = \text{softmax}(x_{target}h_s^T)h_s = c_s, \tag{9}$$

where $c_s$ denotes the context vector obtained by upsampling $h_s$ to match the length of the mel-spectrogram $x_{target}$. As shown in Figure 3(a), the product $x_{target}h_s$ forms $A_{log}$, which serves as the target for the $v_{SAG}$. We define a vector field from $A_{hard}$ to $A_{log}$ via the following ordinary differential equation (ODE):

$$u_t^{SAG} = A_1 - (1 - \sigma_{min})A_0, \tag{10}$$

where $A_1$ and $A_0$ denote $A_{log}$ and $A_{hard} + \epsilon$, respectively. The hard alignment matrix $A_{hard}$ is obtained from $A_{log}$ using monotonic alignment search Kim et al. (2020). The network $v_{SAG}$ is trained to predict the vector field by minimizing the $\mathcal{L}_{SAG}$ between the predicted vector field $v_{SAG}(A_t, h_s, t; \theta_{SAG})$ and the target vector field $u_t$:

$$\mathcal{L}_{SAG} = \left\| u_t^{SAG} - v_{SAG}(A_t, h_s, t; \theta_{SAG}) \right\|_2^2, \left( A_t = tA_1 + (1 - (1 - \sigma_{min})t) A_0 \right) \tag{11}$$

where $\theta_{SAG}$ denotes the learnable parameters of the $v_{SAG}$. During inference, the soft alignment matrix $A_{soft}$ can be obtained by applying $v_{SAG}$ using only $h_s$ and $A_{hard}$, which is predicted from the duration predictor. The vector $h_s$ has 256 dimensions and length $N$ corresponding to the phoneme sequence. It is repeated $T$ times such as Elias et al. (2021a), where $T$ is the length of the mel-spectrogram. Thus, the repeated $h_s$ has a dimension of $N \times T \times 256$ and is concatenated with $A_t$, resulting in a feature of dimension $N \times T \times 257$, which is then fed into $v_{SAG}$. The network then predicts the vector field and generates $A_{soft}$ from $A_{hard}$ by iteratively applying

$$A_{t+dt} = A_t + v_{SAG}(A_t, h_s, t; \theta_{SAG})dt, \tag{12}$$

where $dt = \frac{1}{N_{SAG}}$ and $N_{SAG}$ is the total number of sampling steps for $v_{SAG}$, with $t$ progressing from 0 to 1. $v_{SAG}$ is implemented as a Conv2D-based U-Net[1], with the input channel size set to 257. The number of feature channels in each block is reduced by a factor of 8 compared to the original implementation.

## 3.3 INVERTIBLE ENCODER

As mentioned in subsection 2.2, the invertible encoder models the residual information $z$ between the input and the conditional feature. Therefore, we utilize it to extract acoustic information absent from

---
[1] https://github.com/milesial/Pytorch-UNet

the context vector, by processing the mel-spectrogram. The architecture of the invertible encoder is illustrated in Figure 3(b), which is normalizing flow Kingma & Dhariwal (2018). The normalizing flow takes the mel-spectrogram $x_{target}$ as input and uses the context vector $c_s$ as a conditional feature. Accordingly, we minimize the mutual information between $c_s$ and $z$ as follows:

$$\mathcal{L}_{\text{NF}} = \text{KL}(p(z|c_s)|q(z)), \quad (13)$$

where $q(z)$ denotes the prior distribution of the invertible encoder. By minimizing $\mathcal{L}_{\text{NF}}$, we align $z$ with the prior distribution. When $\mathcal{L}_{\text{NF}}$ is minimized, the mutual information between the residual information $z$ and the context vector $c_s$ is also minimized according to Eq. (5) as follows:

$$I(z, c_s) \leq \mathcal{L}_{\text{NF}}. \quad (14)$$

Consequently, $z$ captures acoustic information that is not contained in $c_s$ but exists in the mel-spectrogram. This approach compensates for the insufficient information present in $c_s$.

During inference, the invertible encoder is not required; instead, residual information can be sampled directly from the prior distribution. We set the prior distribution $q(z)$ to a Gaussian distribution $\mathcal{N}(0, 1)$. Since the decoder of RisoTTo is trained with $z$ distributed according to this Gaussian prior, similar effects can be observed by using Gaussian noise sampled from the prior distribution, as empirically demonstrated in prior numerical work Lee et al. (2022); Li et al. (2025); Lee & Kim (2019); Lee et al. (2020). This is conceptually similar to the use of $z$ in VAEs, where the latent variable carries compressed but informative characteristics of $x_{target}$. However, a known limitation of VAE Kingma et al. (2013) is that the range of information extraced is inherently dependent on dimension of $z$. In contrast, our method effectively models the residual information between $c_s$ and $x_{target}$ through Eq. (13).

### 3.4 PROMPT-AWARE LIGHTWEIGHT CONVOLUTION

Lightweight convolution Wu et al. (2019) employs a fixed context window and reuses the same weights for all context elements, regardless of the current time step. This property can be particularly advantageous for TTS, where relevant context elements tend to be more local compared to tasks such as machine translation or other language processing tasks, as discussed in Elias et al. (2021b). Therefore, we adopt a lightweight convolution block composed of lightweight convolution, a gated linear unit (GLU) Veness et al. (2021), and a feed-forward layer with residual connections as shown in Figure 3(c). To further improve the performance of zero-shot TTS, we extend the lightweight convolution with a speaker-adaptive mechanism inspired by SC-CNN Yoon et al. (2023). Specifically, SC-CNN utilizes speaker embeddings to modulate the convolutional kernel weights dynamically. Following this approach, the speech prompt encoder receives a speech prompt—a 3-second mel-spectrogram segment extracted from the reference speech—and generates a speaker embedding $s$. This embedding $s$ is then reshaped to serve as the kernel weights of the lightweight convolution. For the text encoder, which uses a $3 \times 1$ lightweight convolution with 8 heads, $s$ is reshaped from a 256-dimensional vector to a $3 \times 8$ matrix. Similarly, when applied to the mel decoder, which employs a $17 \times 1$ lightweight convolution with 8 heads, $s$ is reshaped to $17 \times 8$. Through this process, the lightweight convolution adapts its kernel weights dynamically based on the speech prompt, allowing it to extract local contexts that are tailored to each speaker. This adaptive mechanism contributes to improved zero-shot TTS performance.

### 3.5 LOSS FUNCTION

In this section, we describe the loss functions used to train RisoTTo. The loss $\mathcal{L}_{\text{SAG}}$, defined in Eq. (11), is used to train the SAG network to predict the vector field between the hard alignment $A_{hard}$ and the soft alignment $A_{soft}$. The loss function $\mathcal{L}_{\text{dur}}$ for training the duration predictor is defined as the L2 distance between the target and predicted durations. Target is obtained from $A_{log}$. Thus, the total loss $\mathcal{L}_{total}$ function is described as follows:

$$\mathcal{L}_{total} = \mathcal{L}_{\text{mel}} + \mathcal{L}_{\text{post}} + \mathcal{L}_{\text{dur}} + \lambda \mathcal{L}_{\text{NF}} + \mathcal{L}_{\text{SAG}}, \quad (15)$$

where $\lambda$ is hyper-parameter set to 10.

## 4 EXPERIMENTS

We used LibriTTS-R Koizumi et al. (2023) (580 hours of data from 2,456 speakers), HiFi-TTS Bakhturina et al. (2021) (292 hours of data from 10 speakers), and LJSpeech-1.1 Ito & Johnson

(2017) (24 hours of data from single speaker) datasets such as Kim et al. (2025) for training RisoTTo. We selected 2,450 speakers from our train set (LibriTTS-R, HiFi-TTS, and LJSpeech-1.1). Sampling rate was set to 22,050Hz, and mel-spectrogram was extracted with a hop size of 256 and a window size of 1024. Then, our test set was prepared as out-of-domain and consists of VCTK Christophe et al. (2017) and the Seed-TTS test set Anastassiou et al. (2024), which are widely used for evaluating zero-shot TTS performance. In addition, we employ a NVIDIA RTX 3090 GPU with a batch size of 32, and Adam optimizer Kingma & Ba (2014) is used with a scheduled learning rate same as FastSpeech2 Ren et al. (2020). Finally, we used *g2p-en*[2] to convert the text into phoneme sequence. Also, HiFi-GAN Kong et al. (2020), which is trained with same train set of RisoTTo, was used as vocoder that converts mel-spectrogram into waveform.

**Evaluation metrics:** For fair evaluation, we employed pre-trained NISQA-MOS Mittag et al. (2021) to evaluate naturalness of speech, instead of human evaluation. For objective evaluation, speaker embedding cosine similarity (SECS), representing speaker similarity, was computed using ECAPA-TDNN Desplanques et al. (2020) pre-trained with Voxceleb Nagrani et al. (2017). Also, we evaluated speech intelligibility with word error rate (WER), using a pre-trained speech recognition model from the official implementation of Whisper Radford et al. (2023) to measure transcription errors from generated samples. In the following experiments, the sampling numbers of flow matching in SAG and PostNet are 2 and 5, respectively.

## 4.1 ALIGNMENT MODELING VIA SAG

We compared our soft alignment generation method with hard and Gaussian upsampling approaches to evaluate the effectiveness of the proposed method. The hard upsampling method, used in most NAR models such as Ren et al. (2020); Han et al. (2024), increases the length of the text representation by simply duplicating each phoneme according to its duration. In contrast, the Gaussian upsampling Donahue et al. (2020) performs differentiable upsampling, which can improve the speech likelihood of the TTS model. This is achieved by computing a weighted sum of the text representations to generate the context vector, optimized to minimize the mel-spectrogram loss ($\mathcal{L}_{\text{mel}}$). However, the range of the weighted sum is limited by the variance of the Gaussian distribution, which constrains the flexibility of the filter. Since our method used attention mechanism that conducts weighted sum for whole text representation, which provides more flexibility than Gaussian upsampling. In inference, flow matching, which observes optimized soft alignment about text representation, generates soft alignment without mel-spectrogram.

| Upsampling | MOS(CI) | WER | SECS |
|---|---|---|---|
| Attention | 4.24±0.09 | 4.83 | 0.694 |
| Hard | 3.85±0.05 | 5.11 | 0.649 |
| Gaussian | 4.07±0.11 | 5.37 | 0.672 |
| SAG (ours) | 4.19±0.10 | 5.03 | 0.681 |

Table 1: Zero-shot TTS performance of RisoTTo according to upsampling methods. "CI" represents 95% confidence intervals.

Table 1 shows the results of RisoTTo using a different upsampling module instead of the SAG network. Attention mechanism in Table 1 denotes soft alignment produced from attention mechanism with target mel-spectrogram. For this evaluation, we randomly selected 6 unseen speakers for each upsampling method from the test set and generated 5 utterances per speaker. Gaussian upsampling outperformed hard upsampling, but Attention mechanism provides more delicate soft alignment than it. Thus, SAG network is trained using soft alignment of attention mechanism and shows better performance of Gaussian upsampling.

## 4.2 RESIDUAL MODELING VIA INVERTIBLE ENCODER

In this subsection, we demonstrate that the invertible encoder effectively captures the residual information between the mel-spectrogram and the context vector. This residual information should be disentangled from the context vector. To validate this, we employed Maximum Mean Discrepancy

---

[2]https://github.com/Kyubyong/g2p

(MMD)Gretton et al. (2012), a statistical distance metric that measures the difference between two probability distributions based on their sample means in a reproducing kernel Hilbert space. A lower MMD value indicates greater similarity between the distributions. Table 2 presents the MMD scores between the context vector $c_s$ and the residual variable $z$, comparing the use of the invertible encoder and a variational autoencoder (VAE) for residual modeling. The results show that the $z$ extracted by the invertible encoder exhibits lower statistical similarity with $c_s$ than that of the VAE, indicating better disentanglement. We also computed the MMD between $z$ and samples $\epsilon \sim N(0,1)$ drawn

| Algorithm | $(c_s, z)$ | $(z, \epsilon)$ |
|---|---|---|
| Invertible encoder | 2.613 | 0.207 |
| VAE | 1.941 | 0.611 |

Table 2: The results of MMD score when using invertible encoder and VAE. ($a$,$b$) denotes MMD score between $a$ and $b$. These score was computed using 50 utterances of validation set

from the prior Gaussian distribution. The $z$ from the invertible encoder is closer to the prior distribution than that from the VAE. In contrast, the VAE tends to produce a $z$ that is too close to the prior, leading to posterior collapse—where $z$ becomes uninformative. Therefore, $z$ from the VAE should not be forced to align too closely with the prior distribution. This is a critical distinction between the invertible encoder and the VAE. The invertible encoder extracts a deterministic latent representation $z$, whereas the VAE produces a distribution from which stochastic variable $z$ are sampled. Because the invertible encoder generates deterministic representations, it is inherently free from posterior collapse. As a result, $z$ can be more closely aligned with the prior distribution, which helps reduce the mismatch of $z$ between training and inference phases in the TTS model.

We further demonstrate that incorporating an invertible encoder can enhance the performance of non-autoregressive TTS models. In Table 3, the autoregressive and non-autoregressive baselines refer to Transformer TTS Li et al. (2019) and FastSpeech, which used linear layer-based feed forward layer, respectively. All models in Table 3 were trained on the LJSpeech-1.1 dataset Ito & Johnson (2017), which contains 13,100 utterances recorded by a single female speaker. We used 12,000 utterances for training, and 50 each for validation and testing. From each trained model, 15 utterances were generated and evaluated. As presented in Table 3, the use of an invertible encoder leads to a more substantial performance improvement in non-autoregressive TTS compared to the conventional VAE-based approach.

| Model | MOS(CI) | WER |
|---|---|---|
| AR | 3.73±0.11 | 6.93 |
| NAR | 3.38±0.13 | 7.05 |
| NAR w/ IE | 3.64±0.10 | 6.96 |
| NAR w/ VAE | 3.51±0.08 | 7.14 |

Table 3: The results on TTS models trained with LJSpeech-1.1 dataset.

### 4.3 EVALUATION

Recently, many zero-shot TTS models were introduced but the official implementation code is not released. To compare RisoTTo with other models, we adopted two approaches. First, for zero-shot TTS models without publicly available official code, we obtained audio samples from their official demo pages. For a fair comparison, we only used samples generated with the same test set (VCTK) as ours. To the best of our knowledge, VALL-E, NaturalSpeech2, and T5-TTS have released zero-shot TTS samples on their demo pages using VCTK. Second, for zero-shot TTS models with publicly available official code, we fairly used their released pre-trained versions. These models were evaluated on the same test sets as ours, namely VCTK and the Seed-TTS test set. Consequently, we selected Spark-TTS, MaskGCT, and F5-TTS, which have publicly released official code. We generated a total of 60 audio samples on VCTK using zero-shot TTS with speech prompts from 10 randomly selected speakers. For the Seed-TTS test set, we created 180 audio samples from 30 randomly chosen speech prompts and conducted the evaluation. VALL-E Wang et al. (2023) is one of the most well-known zero-shot TTS models, introducing a decoder-only architecture to model

| Model | Latency | #Param. | VCTK | | | Seed-TTS testset | | |
|---|---|---|---|---|---|---|---|---|
| | | | MOS(CI) | SECS | WER | MOS(CI) | SECS | WER |
| GT | - | - | 4.42±0.09 | 0.851 | 3.81 | 4.14±0.11 | 0.825 | 4.41 |
| VALL-E* | 6.4s | 302M | 3.92±0.10 | 0.541 | 6.42 | n/a | n/a | n/a |
| T5-TTS* | 5.6s | 220M | **4.21±0.09** | 0.613 | **4.91** | n/a | n/a | n/a |
| NaturalSpeech2* | 3.6s | 435M | 3.71±0.22 | 0.587 | 5.52 | n/a | n/a | n/a |
| Spark-TTS | 6.86s | 500M | 3.93±0.12 | 0.623 | 6.19 | 3.82±0.13 | 0.616 | 6.73 |
| F5-TTS | 4.24s | 336M | 4.02±0.15 | 0.646 | 5.86 | 3.95±0.11 | 0.632 | 6.18 |
| MaskGCT | 5.74s | 1048M | 4.18±0.11 | 0.637 | 5.37 | **4.01±0.12** | 0.618 | 5.94 |
| RisoTTo | 0.89s | 33M | 4.14±0.13 | **0.668** | 5.51 | 3.92±0.08 | **0.651** | **5.63** |

Table 4: The results on zero-shot TTS using Seed-TTS testset. GT denotes ground truth of waveform. The asterisk (*) indicates that the measurement was taken from samples on the official demo page. Latency and #Param. denote the time required to generate a 10-second speech sample and the number of model parameters, respectively.

| Model | MOS(CI) | SECS | WER |
|---|---|---|---|
| GT | 4.47±0.07 | 0.851 | 3.61 |
| RisoTTo | 4.11±0.10 | 0.673 | 5.15 |
| RisoTTo w/o PAL | 4.08±0.12 | 0.638 | 5.22 |
| RisoTTo w/o SAG | 3.96±0.12 | 0.669 | 5.19 |
| RisoTTo w/o IE | 4.01±0.09 | 0.681 | 5.47 |

Table 5: The results on zero-shot TTS using VCTK dataset. Using 10 speakers of VCTK, 50 audio samples were generated.

discrete tokens from a neural audio codec. T5-TTS Neekhara et al. (2024), based on an encoder-decoder architecture, was also included as a representative autoregressive model. NaturalSpeech2 Shen et al. (2023) is non-autoregressive TTS using latent diffusion for generating latent representation of Encodec Défossez et al. (2022). Then, F5-TTS Chen et al. (2024) is a non-autoregressive TTS model based on a flow-matching network composed of DiT Peebles & Xie (2023) blocks. Finally, Spark-TTS and MaskGCT adopt an autoregressive framework and a masked generative codec Transformer architecture, respectively. As shown in Table 4, we compared RisoTTo with the aforementioned models in terms of MOS, SECS, and WER. As a result, our RisoTTo outperforms all models except T5-TTS and MaskGCT, which is remarkable given its relatively small number of parameters. Furthermore, thanks to its lightweight parameterization and non-autoregressive architecture, RisoTTo achieves significantly faster performance compared to existing models. While T5-TTS and MaskGCT outperform RisoTTo in terms of MOS and WER, RisoTTo exhibits a higher SECS.

## 4.4 ABLATION STUDY

In this section, we compare the effects of the three proposed modules on zero-shot TTS performance. As shown in Table 5, the invertible encoder has a significant impact on speech quality, since removing it leads to a substantial degradation in speech intelligibility. In the case of the prompt-aware lightweight convolution, its influence on speech quality is relatively minor, but it plays an important role in maintaining speaker similarity. Finally, the soft alignment generation module has a considerable effect on MOS.

## 5 CONCLUSION

We proposed three algorithm that can improve performance of zero-shot non-autoregressive TTS. Soft alignment generation upsamples text representation to richer context vector, and invertible encoder effectively models residual information about acoustic representation. Then, prompt-aware lightweight convolution enhances speaker similarity via kernel weight depending on speech prompt. Thus, RisoTTo achieved better performance compared with representative zero-shot autoregressive TTS. Our future work considers improvement of latency and more diverse comparisons to evaluate performance.

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

# A APPENDIX

## A.1 THE USE OF LARGE LANGUAGE MODELS

Large Language Models were used only for minor language editing and grammar correction. No part of the research design, analysis, or scientific writing relied on LLMs.

