# OpenReview forum: "Zero-Shot Non-Autoregressive TTS Beyond Autoregressive Models Using Soft Alignment Generation and Residual Modeling"
_ICLR.cc/2026/Conference — Submitted to ICLR 2026_

### Official Review · Reviewer_D4GQ · 2025-10-18

**Soundness:** 1
**Presentation:** 1
**Contribution:** 2
**Rating:** 2
**Confidence:** 5

**Summary:**

The author contributes by proposing methods to enhance non-autoregressive (NAR) text-to-speech (TTS) models:
1) They introduce a technique to distill soft alignments from autoregressive models into a flow matching model, enabling non-autoregressive models to use attention-like context vectors.
2) They design an invertible encoder using normalizing flow to separate semantic and acoustic representations.
3) They propose a prompt-aware lightweight convolution to improve zero-shot TTS performance.

**Strengths:**

- This approach of distilling soft alignments from autoregressive models into a flow-matching model to enable attention-like context vectors in non-autoregressive models is an intriguing concept.

**Weaknesses:**

**[W1]** The evaluation in this paper has significant issues that make the experimental results completely unconvincing. The authors did not use mainstream practices:
1) LibriSpeech test-clean 1234 samples (VALL-E release);
2) full set of Seed-TTS test-EN/ZH/Hard (MaskGCT, F5-TTS, CosyVoice 1/2/3);

Instead, they cherry-picked the Seed-TTS test set, creating 180 audio samples from 30 randomly selected speech prompts for evaluation. The WER values of the baselines (Spart-TTS, F5-TTS, MaskGCT) in Table 4 are much worse than the values reported in the original papers.

Additionally, the authors obtained audio samples from their official demo pages for comparison, which is highly unusual (it's the first time I've seen such an approach) and unreliable.

---
**[W2]** The majority of the references in the author's work are from three years ago, with only one being from 2025. The author’s approach to NAR's zero-shot TTS model is still based on the encoder-decoder architecture (Transformer-based, T5-based), which has already become outdated in the current era of decode-only models. Therefore, I question the timeliness of this work.

**Questions:**

The following are the revision suggestions:

1) Redo the experiments. For zero-shot TTS, use LibriTTS for training with small data, and use Libriheavy or Emilia for experiments with large data.

2) Redo the evaluation. For the test set in English, use: LibriSpeech test-clean 1234 samples & Seed-TTS test-EN full set with 1000+ samples

3) Clearly report the ASR model variant used for WER, such as the most widely used Whisper Large v3, to avoid ambiguity.

4) Subjective experiments should report more details, and it is recommended to include them in the appendix, such as the number of participants, their native language background, instructions, scoring tables, etc.

5) Add efficiency metrics, including RTF.

6) For the baseline, use SOTA methods such as NAR F5-TTS, ZipVoice, MaskGCT for non-autoregressive (NAR) models, and CosyVoice 2 for autoregressive (AR) models. Using the numbers from the original papers will make the comparison more convincing.

---

### Official Review · Reviewer_NUUy · 2025-10-31

**Soundness:** 1
**Presentation:** 2
**Contribution:** 2
**Rating:** 2
**Confidence:** 4

**Summary:**

This paper introduces **RisoTTo**, a zero-shot non-autoregressive (NAR) text-to-speech (TTS) model that attempts to transfer the benefits of autoregressive models through three main components:
1. **Soft Alignment Generation (SAG)** based on flow matching,
2. **Invertible Encoder (IE)** for residual acoustic modeling, and
3. **Prompt-Aware Lightweight Convolution (PAL)** conditioned on a 3-second speech prompt.

The authors claim that this approach achieves comparable quality to large autoregressive (AR) systems while maintaining faster inference and fewer parameters.

**Strengths:**

- The topic — improving zero-shot NAR TTS — is relevant and timely.
- The integration of existing techniques (flow matching, invertible flows, lightweight convolution) might have some limited engineering value.

**Weaknesses:**

### **1. Lack of real novelty**
- The paper merely combines known ideas (flow matching, normalizing flow, speaker-conditioned convolution) without introducing new theory or insight.
- It ignores recent, stronger NAR paradigms such as diffusion- or consistency-based TTS (e.g., F5-TTS, Ditto-TTS).

### **2. Unconvincing and inconsistent design**
- The model still relies on many modules and losses (SAG, invertible encoder, duration predictor, post-net, etc.), which contradicts the goal of scalability.
- The 3-second *fixed-length* prompt embedding limits speaker modeling capacity.
- The duration predictor is trained only with an **L2 loss**, producing flat prosody and unnatural rhythm.
- The “prompt-aware convolution” is poorly justified and shows negligible improvement.

### **3. Questionable experimental credibility**
- **No demo samples** are provided, which is unacceptable for a TTS paper.
- **Table 4** is highly suspicious — latency and SECS values are reported for models (e.g., VALL-E, NaturalSpeech2, T5-TTS) that have **no public code**. It is unclear how these results were obtained.
- The model allegedly outperforms systems with 10× more parameters and training data, which is implausible without human evaluation.
- The results rely solely on automatic MOS (NISQA), which cannot replace human perceptual tests.

### **4. Weak evaluation and analysis**
- The zero-shot setting is not convincingly tested — no cross-lingual, unseen-speaker, or unseen-domain evidence is provided.
- Ablation studies are minimal and do not explain *why* each module helps.

### **5. Poor reproducibility and transparency**
- The paper does not release code, training details, or implementation settings.
- Multiple flow-based components and loss functions make the approach unstable and difficult to reproduce.
- Some reported results (e.g., SECS across tables) are inconsistent and lack statistical validation.

**Questions:**

1. How were latency and SECS values for VALL-E, NaturalSpeech2, and T5-TTS obtained?
2. Why are no audio samples or demos provided?
3. How does the model compare against modern diffusion or consistency-based NAR TTS (e.g., F5-TTS, Ditto-TTS)?
4. Why use a fixed 3-second prompt and static embedding instead of variable-length conditioning?
5. How sensitive is performance to the weighting of multiple losses (Lmel, Lpost, Ldur, LNF, LSAG)?
6. Can the authors release implementation or evaluation code for verification?

---

### Official Review · Reviewer_fcTM · 2025-10-31

**Soundness:** 2
**Presentation:** 2
**Contribution:** 2
**Rating:** 4
**Confidence:** 3

**Summary:**

This paper presents RisoTTo, a non-autoregressive (NAR) model for zero-shot text-to-speech (TTS) that aims to incorporate the benefits of autoregressive (AR) models. The authors propose a collection of techniques to bridge the performance gap between NAR and AR systems. These include: 1) a Soft Alignment Generation (SAG) module, based on flow matching, to distill soft alignments from AR attention mechanisms for use in a NAR setting; 2) an Invertible Encoder (IE) based on normalizing flows to model the residual acoustic information not captured by the text-aligned context vector; and 3) a Prompt-Aware Lightweight Convolution (PAL) to improve speaker adaptation. The paper presents experimental results showing that the proposed model achieves performance comparable to several state-of-the-art AR and NAR zero-shot TTS systems, particularly in terms of speaker similarity.

**Strengths:**

The paper tackles the important and challenging problem of improving non-autoregressive TTS by emulating the strengths of autoregressive models. The core ideas are technically interesting; using flow matching to learn a soft alignment mechanism is a novel approach, and the use of an invertible encoder to explicitly model residual information is a conceptually sound way to enrich the decoder's input. The Prompt-Aware Lightweight Convolution is also a clever, efficient mechanism for speaker conditioning. The experimental results, particularly the high speaker embedding cosine similarity (SECS) scores, are a clear strength, suggesting that the model is very effective at capturing and reproducing the target speaker's voice characteristics.

**Weaknesses:**

My main concern with this work is its overall complexity and fragmented nature. The proposed model is an intricate assembly of many distinct, sophisticated components (flow matching for alignment, a separate invertible encoder for residuals, a specific prompt-aware convolution, a post-net, etc.). This design philosophy feels somewhat contrary to the current trend in the field, which is moving towards more unified, end-to-end, and scalable architectures.

Furthermore, while the combination of these techniques yields good results, the central innovation seems to be quite incremental. The idea of using an invertible encoder or VAE to model residual or stochastic information between text and speech is not entirely new and bears a strong resemblance to prior work. For instance, the general architecture is reminiscent of approaches proposed in papers such as "VARA-TTS: Non-Autoregressive Text-to-Speech Synthesis based on Very Deep VAE with Residual Attention" and "VAENAR-TTS: Variational Auto-Encoder based Non-AutoRegressive Text-to-Speech Synthesis," which also use VAEs to model this information gap in NAR TTS. The paper would be strengthened by acknowledging and differentiating itself from these highly relevant prior works.

**Questions:**

I have a few questions that I hope the authors can address to help clarify their contribution:
1. The model's architecture is quite complex, integrating multiple distinct generative modeling paradigms. Could you comment on the scalability of this "fragmented" approach? Do you see a path to simplifying this design while retaining its benefits, in line with the field's move towards more unified models?
2. The central contribution appears to be the use of an invertible encoder to model the residual information between the mel-spectrogram and the text-aligned context vector. This is a very subtle but important component. How does this approach fundamentally differ from prior works, such as "VARA-TTS: Non-Autoregressive Text-to-Speech Synthesis based on Very Deep VAE with Residual Attention" and "VAENAR-TTS: Variational Auto-Encoder based Non-AutoRegressive Text-to-Speech Synthesis," which also use a VAE-based structure to model latent variables for NAR TTS? A direct comparison or discussion would be very insightful.
3. Given the complexity, could you provide a more detailed ablation or analysis to show which of the three proposed components (SAG, IE, PAL) is most critical for the performance gains? For example, what is the performance if you only use the Invertible Encoder on top of a standard NAR baseline with duration-based upsampling?

---

### Official Review · Reviewer_63E9 · 2025-11-01

**Soundness:** 2
**Presentation:** 3
**Contribution:** 2
**Rating:** 2
**Confidence:** 4

**Summary:**

This paper presents RisoTTo, a Zero-Shot Non-Autoregressive Text-to-Speech (TTS) system, that aims to bridge the performance gap between non-autoregressive (NAR) and autoregressive (AR) models with the following fetures:

Soft Alignment Generation (SAG) is a flow-matching model that learns soft text–mel alignments distilling the AR attention behavior.

Invertible Encoder (IE) is a normalizing flow–based encoder that disentangles semantic and acoustic residual representations, enabling Gaussian sampling during inference.

Prompt-Aware Lightweight Convolution (PAL) are  dynamically modulated convolutional kernels conditioned on the speaker prompt for improved zero-shot adaptation.

Extensive experiments across multiple datasets (LibriTTS-R, HiFi-TTS, LJSpeech, VCTK, Seed-TTS) show that RisoTTo achieves competitive MOS, SECS, and WER compared to strong AR baselines (T5-TTS, MaskGCT, VALL-E) while being significantly more efficient (33M parameters, ~0.89s latency per 10s audio).

Ablation studies verify that IE primarily improves intelligibility, SAG enhances naturalness, and PAL increases speaker similarity.

**Strengths:**

Well-motivated problem: Addressing the quality gap between NAR and AR zero-shot TTS is an important and active research direction.

Elegant architecture: The SAG, IE, and PAL modules complement each other to improve contextual representation, residual modeling, and speaker conditioning.

Efficiency: Very lightweight model with low inference latency, which is valuable for deployment.

**Weaknesses:**

Numerous inconsistencies and apparent errors exist in the relevant experimental results and experimental design.

Experimental Reliability and Validity:
- The latency and WER numbers reported for _VALL-E_, _T5-TTS_, and _NaturalSpeech2_ in Table 4 are fundamentally unreliable because they are derived from **official demo samples** rather than controlled, reproducible experiments.
- Demo pages do not provide inference-time measurements, and audio files are typically pre-rendered and post-processed, meaning that latency cannot be accurately derived from them.
- It is unclear how the authors computed latency for these baselines, as official demos do not expose real-time generation logs, model configuration, or hardware conditions.
- Without identical inference hardware, batch size, and pipeline settings (text frontend, vocoder, postnet, etc.), comparing latency values is meaningless.
- For  WER demo pages contain very few utterances—often <20 per model—and the spoken texts differ across demos. This violates any controlled variable condition required for fair metric comparison.
- The mismatch in phonetic content and text length across models directly biases WER values. For example, longer sentences or punctuation differences can inflate WER independently of model quality.

Implausibly Strong Results Given the Small Model Size and Limited Data:
- The reported zero-shot performance of RisoTTo appears unrealistically strong considering its training scale and model size.
- The paper claims to train on a few open datasets (LibriTTS-R, HiFi-TTS, LJSpeech) and a model with ~33M parameters.
- Table 4 shows RisoTTo achieving comparable or superior MOS and SECS scores to systems like F5-TTS and Spark-TTS, which are trained on hundreds of thousands of hours of proprietary or large-scale multi-speaker data.
- Such a large performance gap—without proportional training data—raises questions about the evaluation setup, the correctness of metric computation, or potential data leakage.
- Without a fair data and model size comparison, the claim that RisoTTo “matches large-scale zero-shot TTS models” lacks credibility.
- The claimed zero-shot performance seems implausible given the small model scale and modest training data. Either additional large-scale pre-training was used (which must be disclosed), or the reported numbers are over-optimistic due to limited and uncontrolled evaluation.

Lack of Controlled Experimental Setup
- Table 4 mixes results from different evaluation sources (demo pages, papers, and authors’ experiments
- The texts, speakers, sampling rates, and vocoders are inconsistent across systems.
- Therefore, any claim of superiority in MOS, SECS, or WER cannot be scientifically substantiated.
- The absence of a controlled, unified test set invalidates the quantitative comparison. The authors should rerun all baselines on the same zero-shot evaluation corpus using released checkpoints and report consistent metrics.

The experimental results in Table 4 appear methodologically flawed. Latency values for demo-based models are meaningless; WER metrics are computed from non-aligned texts and insufficient sample size; and RisoTTo’s performance claims seem inconsistent with its limited training scale. Without controlled evaluation or transparent methodology, Table 4 undermines the credibility of the entire empirical section. The authors must either (1) rerun fair baseline comparisons under identical conditions or (2) remove the questionable numbers from the paper.

The paper does not provide any demo page or audio samples, which is a major omission for a TTS paper. For any text-to-speech work, especially those claiming improvements in naturalness, speaker similarity, and zero-shot generalization, it is essential to allow reviewers and readers to **listen to generated samples**. The entire evaluation of perceptual quality (MOS, SECS, WER) depends on audio output quality — yet the paper does not share any listening evidence or demonstration website.

Moreover, since Table 4 comparisons rely on demo samples from other models (VALL-E, T5-TTS, etc.), **it is inconsistent and unfair** that the proposed method itself provides no equivalent demo for verification. This makes the evaluation **non-transparent and non-reproducible**, undermining the credibility of the claimed superiority in Table 4 and Table 5.

The absence of a demo page is a serious flaw for a speech synthesis paper. The authors must provide listening examples of their model’s output—ideally aligned with the same evaluation texts used for WER and MOS calculations—to enable fair and transparent assessment. Without such evidence, it is impossible to judge the claimed improvements in perceptual quality.

**Questions:**

How was the weighting factor λ=10 chosen? The total loss combines heterogeneous objectives (mel distance, flow loss, KL loss) with different scales. Without sensitivity analysis, optimization stability is questionable.

The paper reports latency and WER for VALL-E, T5-TTS, and NaturalSpeech2 using official demo samples. How were these numbers computed if demo pages do not provide inference logs, and texts differ across samples?

Why does the paper not include a demo page for listening verification? In TTS research, reviewers must hear the generated audio to judge perceptual quality. Without demos, the automatic MOS values cannot be trusted.

The paper claims the method “goes beyond autoregressive models,” yet there is no formal analysis showing that the NAR model is more expressive or achieves comparable likelihoods. Could you provide a theoretical justification or empirical evidence for this claim? Without a probabilistic comparison, the “beyond AR” statement reads as marketing rather than a substantiated claim.

---

### Meta-Review · Area_Chair_HXXg · 2025-12-25

**Summary:**

The paper aims to improve non-autoregressive TTS with improvements on the alignments, the separation of acoustic and content information, and the conditioning on the prompts. The alignments are based on an learning an alignment matrix using flow matching. The separation of acoustic and content information is based on learning residual information using an invertible network. Finally, the conditioning of speech prompts is based on a lightweight CNN. Latency and MOS on VCTK and the Seed-TTS test set are used for evaluation. There are a few experiments justifying the design choices.

**Reviewer Concerns:**

The main concern among reviewers is the evaluation. Reviewer 63E9, NUUy, and D4GQ question how latency numbers are collected given that many models compared do not come with an official implementation.

Novelty is the second concern raised by reviewer NUUy and D4GQ. Coupled with the novelty is how the paper handles the many moving parts, and reviewers NUUy and fcTM raise reproducibility concerns.

**Reviewer Scores:**

No rebuttals are submitted.

---

### Decision · Program_Chairs · 2026-01-26

Reject